# Plant Gel-Mediated Synthesis of Gold-Coated Nanoceria Using *Ferula gummosa*: Characterization and Estimation of Its Cellular Toxicity toward Breast Cancer Cell Lines

**DOI:** 10.3390/jfb14070332

**Published:** 2023-06-21

**Authors:** Seyed Mousa Mousavi-Kouhi, Abdollah Beyk-Khormizi, Mohammad Sadegh Amiri, Mohammad Mashreghi, Alireza Hashemzadeh, Vahideh Mohammadzadeh, Fariba Alavi, Javad Mottaghipisheh, Mohammad Reza Sarafraz Ardakani, Mohammad Ehsan Taghavizadeh Yazdi

**Affiliations:** 1Department of Biology, Faculty of Sciences, University of Birjand, Birjand 9717434765, Iran; smmousavi@birjand.ac.ir; 2Department of Biology, Faculty of Sciences, Yazd University, Yazd 8915818411, Iran; abdollahbeyk@gmail.com; 3Department of Biology, Payame Noor University, Tehran 19395-4697, Iran; m.s_amiri@pnu.ac.ir (M.S.A.); fariba_al62@yahoo.com (F.A.); 4Department of Medical Biotechnology and Nanotechnology, Faculty of Medicine, Mashhad University of Medical Sciences, Mashhad 91778, Iran; mmashreghi3@gmail.com; 5Nanotechnology Research Center, Pharmaceutical Technology Institute, Mashhad University of Medical Sciences, Mashhad 91778, Iran; 6Department of Pharmaceutical Nanotechnology, School of Pharmacy, Mashhad University of Medical Sciences, Mashhad 91778, Iran; hashemzadehalireza@gmail.com (A.H.); vahide581@gmail.com (V.M.); 7Department of Aquatic Sciences and Assessment, Swedish University of Agricultural Sciences (SLU), SE-750 07 Uppsala, Sweden; 8Applied Biomedical Research Center, Mashhad University of Medical Sciences, Mashhad 91778, Iran

**Keywords:** *Ferula*, gold coating, nanoceria, green synthesis, cell toxicity

## Abstract

In this study, a novel method using *Ferula gummosa* gums as a capping agent was used to synthesize the nanoceria for the first time. The method was economical and performed at room temperature. Furthermore, it was coated with gold (Au/nanoceria) and fully characterized using X-ray powder diffraction (XRD), field emission scanning electron microscopy with energy-dispersive X-ray spectroscopy (FESEM-EDX), Fourier-transform infrared spectroscopy (FTIR), dynamic light scattering (DLS), and zeta potential (ζ potential). The crystallite size obtained from the results was 28.09 nm for Au/nanoceria. The energy-dispersive X-ray spectroscopy (EDX) analysis of Au/nanoceria revealed the compositional constituents of the product, which display the purity of the Au/nanoceria. The cell toxicity properties of the non-doped and Au-coated nanoceria were identified by a MTT analysis on a breast cancer cell line (MCF7). Additionally, human foreskin fibroblast cells (HFF) were used as a normal cell line. The cytotoxicity results indicated that the toxicological effect of Au/nanoceria on cancer cells was significant while having little toxic effect on normal cells. The toxicity effect of nanoceria clearly shows the dependence on dose and time, so, with increasing the dose of Au/nanoceria, the death of cancer cells also increases.

## 1. Introduction

The biomedical properties of nanoceria have attracted great attention to potential biomedical applications [1,2]. The most distinct attribute of nanoceria is the changeable redox behavior of the cerium (Ce^3+^/Ce^4+^) that is controlled by the external environment [3,4]. Therefore, the interchangeable oxidation state gives the ability to scavenge radicals. It inhibits cancer cells’ growth or bacteria and invasion and can act as a radiation and chemotherapy sensitizer. The important factors that affect the function of nanoceria are particle size, environmental acidity, concentration, and exposure time [5,6,7,8]. Before the preliminary in vivo test, the best scenarios evaluate the cytotoxicity of the prepared nanoceria against normal and abnormal cellular models. The main cause of cell death is the change in the concentration of the free radicals and the oxidative stress in the approximate environment [9,10], although the physical contact of the nanoparticles (NPs) and metal ions leaching from the NPs can be other causes [11,12].

Gold NPs have shown potential pharmaceutical applications as antioxidant, antimicrobial, anticancer, molecular diagnostics, and drug delivery agents [13,14,15]. For example, if combined with antibiotics, it can enhance the antibacterial efficiency comparatively. The toxicity issues of gold nanoparticles are exposure time and being size- and dose-dependent in nature [16,17,18]. The lower the size, the more toxicity was observed.

Combining nanoceria and gold would be a novel approach for enhancing the biomedical effects of the nanoceria, which can have complementary or enhancing effects on its biological characteristics [19]. Using this combination in biomedical applications can be of great value in the synthesis of new nanocomposites. It appears that gold-coated nanoceria has a better antioxidant activity and biocompatibility compared to nanoceria alone [20]. However, there is still little knowledge about the toxicity, anticancer, and antibacterial properties of Au-coated nanoceria. Oxidant activity in Au/nanoceria makes it a good candidate for antineoplastic application. Gold acted as a modifier that increased the reactive oxygen species (ROS) and caused mitochondrial damage to the A549 lung cancer cells [19].

Bio-based nanoparticle synthesis offers various benefits over chemical and physical approaches [21,22]. Biological approaches rely on renewable and eco-friendly natural sources such as plant extracts, microbes, and fungus, which are less expensive than chemical and physical approaches, since they do not necessitate complicated equipment, high-pressure conditions, or poisonous compounds [23,24]. It is also nontoxic and safe, eliminating the risks associated with the use of harmful chemicals and expensive equipment. Biosynthesis results in nanoparticles that are more stable, homogenous, and biocompatible in nature, exhibiting high efficiency while maintaining a narrow size range [25,26]. It allows for the creation of nanoparticles with unique functions such as high antibacterial, anticancer, and anti-inflammatory characteristics. Because of these benefits, the biological production of nanoparticles is gaining popularity in a variety of biotechnology and medicinal applications [27,28,29,30].

The family Apiaceae (Umbelliferae) comprises ca. 442 genera and 3575 species/infraspecific taxa in the world [31]. Among the genera of this family, *Ferula* L. is one of the largest, includes many species commonly used in traditional medicine, and is a promising source of biologically active ingredients. Geographically, the c. 180 species of this Eurasian genus are distributed from the Canary Islands in the west throughout the Mediterranean region, Middle East and Central Asia to Western China in the east, and Northern India in the south. The majority of the *Ferula* plants have a pungent odor and can be used for different purposes [32,33,34]. Several species of the genus *Ferula* have gained fame owing to their potential in producing gum, which has many properties in medicine. Among them, *Ferula* gummosa Boiss., *F. assa-foetida* L., *F. latisecta* Rech.f. & Aellen, and *F. foetida* Regel are the most popular species to supply the gum in the global market [35].

Plant gums come from diverse fragments of plants and have a wide array of uses in pharmacology and many other industries [36,37]. The source of gums can be the seed, epidermis, leaves, and bark [38,39,40]. The gum of the most important species of the genus *Ferula*, namely *F. gummosa*, has many therapeutic properties. This gum, known as galbanum, is collected mostly in Iran and Afghanistan and exported from these areas to other countries. In different European countries, galbanum has been used to treat epilepsy, stomachache, and as an effective wound-healing agent. This material has also been used as an anthelmintic agent and to treat diarrhea, constipation, and abdominal pains [32]. In Iranian traditional medicine (ITM), this gum is also known as “Ghasni or Barijeh” and is reported to have numerous medicinal properties. It has been recognized as a sedative and to strengthen the memory. Galbanum has been recommended as a tonic, anticonvulsant, anti-hysteric, and decongestant, and it is useful in the treatment of neurological disorders [41,42]. Moreover, it has been widely prescribed as an appetizer, anthelmintic, emmenagogue, and to treat digestive diseases such as colic and flatulence. It is also reported as an anti-catarrhal, antiallergic, and a good remedy for dyspepsia [43,44].

In summary, gold-coated nanoceria and other bimetallic nanocomposites can open up new avenues in cancer treatment. These materials have unique features such as improved stability and anticancer potency by mixing various metals, which can kill cancer cells by a variety of ways, including oxidative stress and the intracellular transport of therapeutic chemicals. As a result, they might be important candidates for cancer therapy. These particles can be utilized to deliver medications to cancer cells, boosting efficacy, and exhibit synergistic effects while reducing negative effects. Therefore, gold coating can improve the particles’ stability and targeting abilities. More study is needed to fully understand these materials’ potential in cancer treatment, and gold-coated nanoceria and other bimetallic nanocomposites can provide intriguing new possibilities for drug delivery and cancer therapy. Herein, *Ferula gummosa* gum is used as a capping agent to synthesize nanoceria for the first time. The synthetic method is economic and novel, and it is performed at room temperature. Furthermore, the surface of nanoceria is coated with gold (Au/nanoceria) that is fully characterized and used for cell toxicity effects against breast cancer cell lines (MCF7) and normal cells (HFF).

## 2. Results and Discussion

### 2.1. X-ray Powder Diffraction (XRD)

The XRD patterns showed the successful synthesis of nanoceria (Figure 1), which was compatible with reference code 00-004-0593, the cubic crystal system, space group Fm3m, and space group number 225. The experimental and calculated 2theta and d-spacing values were consistent with each other (Table 1). The results indicated the gold deposition on the surface by chemical reduction maintained the nanoceria structural network. The crystallite size obtained from the results was 28.09 nm for the Au/nanoceria. Unfortunately, the values of Au in Au/nanoceria due to the low concentration of the gold did not appear in the XRD pattern. Therefore, we can only investigate the XRD peaks of the nanoceria.

### 2.2. Fourier-Transform Infrared Spectroscopy (FTIR)

The peaks in the plant extract clarified the presence of different functional groups in the plant extract that could interact with the cerium ions. The adsorption peaks at 3400 cm^−1^ were associated with the O-H signal (Figure 2, red). The band 2922 at cm^−1^ was related to C-H stretching. The vibrations of the aromatic C=C bond were observed at 1613 cm^−1^. Moreover, the peaks at approximately 1377 cm^−1^ and 1071 cm^−1^ represented C-H bending and C-O stretching of primary alcohols, respectively. The comparison between the FTIR spectrum of *F. gummosa* and Au/nanoceria revealed the presence of *F. gummosa* in the NPs. The 485 cm^−1^ adsorption band was also related to the Ce-O vibration and the formation of nanoceria [45].

### 2.3. Field Emission Scanning Electron Microscopy (FESEM)

The FESEM images were used to investigate the morphology and the size of the Au/nanoceria (Figure 3a,b,d). The spherical and semispherical morphologies were observed in the solid phase. The particle sizes demonstrated a narrow distribution, and a mean particle size of 31.74 ± 8.64 nm (N = 354, Figure 3d) was obtained using ImageJ software (version 1.53 K). The frequency analysis was performed using IBM SPSS statistics. The comparison of the mean particle size with crystallite showed agglomeration or aggregation, although aggregates with a size of nearly 80 nm also indicated the tendency of the particle to clump together to form relatively larger particles. The energy-dispersive X-ray spectroscopy (EDX) analysis of Au/nanoceria revealed the compositional constituents of the product, that display the purity of the NPs (Figure 3c).

### 2.4. Transmission Electron Microscopy (TEM)

The TEM images also demonstrated spherical and semi-spherical morphologies. The particle size distribution (PSD) was in the range of 5–35 nm, with a mean diameter of 15.76 ± 5.46 nm. The particles were enveloped with spheres that could be composed of organic residues from *F. gummosa* (Figure 4).

### 2.5. Dynamic Light Scattering (DLS) and Zeta Potential (ζ Potential)

The DLS analysis demonstrated a Z average of 139.36 nm with an acceptable polydispersity index (PI = 0.27, Figure 5). The PI showed an acceptable range of hydrodynamic sizes of the particles. A higher Z average compared to the solid phase and crystallite sizes indicated that particles tended to agglomerate in water. The agglomeration was relatively high, and the particle diameter was more than four-times greater than the sizes obtained from the FESEM. It is worth noting that DLS is less sensitive to smaller particles, and the presence of larger particles may have interfered with the results. The zeta potential also displayed a negative surface charge of the Au/nanoceria (−18.1 mV), which may be the logical reason for NP stability in the aqueous solution, along with the particle sizes. It appears that the biosynthesis method at room temperature could influence the particle sizes of the nanoceria.

### 2.6. Cell Toxicity Properties of Biosynthesized Nanocomposites

Nanotoxicology is a part of nanosciences that considers the side effects of nanomaterials [46,47,48,49]. With increasing biological applications, concerns about the safety and toxicity of materials on the cells of living organisms, especially humans, have increased [50,51,52,53,54,55,56]. The size, surface charge, and concentration of nanomaterials are some of the factors that can affect the toxicity of nanoparticles. The MTT examination is a measurable cell toxicity analysis. MTT dye was applied in this analysis. The MTT analysis is a sensitive and usable index to specify the cell metabolic properties [57]. The cytotoxic properties of the fabricated non-doped and Au-coated nanoceria were identified by an MTT analysis on a breast cancer cell line (MCF7). Additionally, HFF (human foreskin fibroblast cells) was used as a normal cell line. The toxicity effect of different concentrations (15.62–500 µg/mL) of biosynthesized nanocomposite was tested at different times (24, 48, and 72 h). The cytotoxicity results of Au/nanoceria are presented in Figure 6. The results of the MTT test indicated that the toxicological effect of Au/nanoceria on cancer cells was significant while having little toxic effect on normal cells. The toxicity effect of Au/nanoceria clearly shows a dependence on dose and time, so, with increasing the dose of Au/nanoceria, the death of cancer cells also increases. Hence, biosynthesized Au/nanoceria can be employed in cancer therapy and drug delivery sciences. Nanoceria are very attractive as possible anticancer NPs because of the exclusive chemistry of nanoceria [58]. The synthesis manner and size of nanoceria affects the toxicity and effectiveness of nanoceria [59]. The smaller size of nanoceria leads to a decreased toxicity of NPs and can be described using the greater tendency of smaller NPs to form agglomerates [60]. There have been several reports that indicated a little toxicity of nanoceria against eukaryotic cells, while no protection efficacy was detected [61,62]. For instance, Gaiser et al. confirmed that, although they adhered to the cell membrane and move in the cells, nanoceria that were smaller than 25 nm or had a size between 1 and 5 μm did not display any substantial cell toxicity [63]. Several scientific sources described a time-/dose-dependent toxicity of nanoceria. In a work, the one-day and ten-day efficacy of nanoceria exposure on diverse cell lines were studied, and the results showed the toxicity tests were significantly dependent on the systems being studied. It did not observe any toxicity results of the nanoceria after 24 h of exposure but reported a genotoxic result for all cell lines after 10 days [64]. Nanoceria mimics ROS-related enzymes that keep normal cells at a physiological pH from oxidative stress and generate ROS in the somewhat acidic tumor microenvironment to cause cancerous cell death [58]. AuNPs are small in size and can enter extensively and be deposited on the tumor situate, bind to many biomolecules and drugs, target drug delivery, and have worthy biocompatibility [14,65]. The use of two nanoparticles that have different applications in cancer treatment can lead to positive results in cancer treatment.

## 3. Materials and Methods

### 3.1. Instruments and Materials

The Bruker D8 Advance was used for the collection of powder X-ray diffraction. The solid-phase images were taken by field emission scanning electron microscopy with energy-dispersive X-ray spectroscopy (FESEM-EDX) using a TESCAN-MIRA3 device. The Fourier-transform infrared spectroscopy (FTIR) was recorded by a Shimadzu 8400. The measurements of dynamic light scattering (DLS) and zeta potential (ζ potential) were done by a particle size analyzer and zeta compact CAD. All the materials were purchased from Sigma and Merck chemical groups unless otherwise stated. The FESEM examination will only work if the samples are prepared suitably. Because metals already conduct electricity when bombarded with electrons, they do not require any preparation [66]. The sample is provided with a gold thin layer. With the use of argon gas and an electric field, gold is attained. An electron from the argon is removed by the electric field, which gives rise to positively charged ions. The negatively charged gold foil attracts these positively charged ions. The argon ions expel gold atoms, which fall onto the sample, covering it with a thin conductive coating. EDX is employed in combination with FESEM. An electron beam with energy of 10–20 keV strikes the conducting specimen’s surface, causing X-rays to emit from the material, and the energy of the emitted X-rays depends on the material under examination. The FTIR examination is employed for the identification of biochemical groups using infrared light for scanning the specimen. Changes in the characteristic pattern of absorption bands obviously show an alteration in the material structure. Radiation from the sources of FTIR devices reach the detector after it passes through the interferometer. The signal is amplified and converted into a digital signal by a convertor and amplifier; after which, the signal is transferred to the computer, where the Fourier-transform is performed. XRD designs could be employed for defining the element amounts if the sample is in a mixture form. The interaction between the X-ray beam and the atomic planes results in partial transmission of the beam, and the rest is absorbed and diffracted by the sample. When nanoparticles charged electrically with enough energy are decelerated, X-rays are created. The produced X-rays are directed at a specimen, which is a finely ground powder. X-rays are identified using the detector, and the signals are done electronically.

### 3.2. Synthesis of Au-Coated Nanoceria (Au/Nanoceria)

*F. gummosa* (0.5 g) in dry form and NaCl (0.9 g) were dissolved in distilled water (100 mL) and stirred for 24 h. Then, Ce(NO_3_)_3_·6H_2_O was dissolved in water (40 mL). The first solution (40 mL) containing plant gums was added to the aqueous solution of Ce(NO_3_)_3_·6H_2_O. After stirring for 24 h, NaOH (0.1) was added dropwise until the nanoceria was formed and the supernatant was clear. Then, it was centrifuged and washed 2 times with 40 mL NaOH and 3 times with distilled water. The precipitate was named nanoceria. In the next step, it was dispersed in an aqueous solution of chloroauric acid (10 mL, 2000 ppm), sonicated briefly, and stirred for 2 h. Then, the mixture was centrifuged, and the supernatant was discarded. The gold ion-coated nanoparticles were lyophilized and powdered. Finally, an aqueous solution of ascorbic acid (3 g, 100 mL) was used to reduce the gold ions to gold. The powder was added continuously and slowly to the ascorbic acid solution under vigorous stirring at room temperature. The Au/nanoceria was separated by centrifuge and washed three times with distilled water to remove any unreacted substrates on the surfaces of the nanoparticles.

### 3.3. Cellular Toxicity Test

The breast cancer cell line MCF7 was purchased from the Pasteur Institute, Iran. The cells were incubated in a culture medium with 10% FBS, 100 μg/mL of streptomycin, and 100 U/mL of penicillin at 37 °C and in a CO_2_ atmosphere. The cellular toxicity properties of bio-fabricated NPs were examined using MTT dye. Concisely, the cell suspension was incubated for 24 h, then treated with the fabricated NPs (15.62, 31.25, 62.50, 125, 250, and 500 μg/mL) for a period of 24, 48, and 72 h. Subsequently, the MTT material was added to all samples, and the plate was incubated at 37 °C. Lastly, purple formazan was added to the wells, and the optical absorbance was measured at 570 nm. The results of the cellular viability were stated as a percentage. Cell viability (%) = (absorbance of sample/absorbance of cells without treatment) × 100. The test was done in triplicate.

### 3.4. Statistical Analysis

Data of the cytotoxicity assay were represented using GraphPad Prism (Version 8). Data are represented as the mean ± SD.

## 4. Conclusions

*F. gummosa* as an important endemic plant of Iran and possesses various industrial and medical indications. Narrow- and small-sized particles of nanoceria were biosynthesized by *F. gummosa* gum at room temperature for the first time. The spherical and evenly distributed particles were of great value and can be used to synthesize the other nanoscale metallic or metal oxides. Gold-coated nanoceria demonstrated good anticancer efficiency, which can be used in the design of new nanocomposites for the treatment of cancer or drug delivery purposes. Gum-mediated fabricated nanocomposites killed cancer cells well while showing a lower toxicity against normal cells compared to the cancer cells. Therefore, we suggest that Au/nanoceria could have more potential and should be tested in more biological applications.

## Figures and Tables

**Figure 1 jfb-14-00332-f001:**
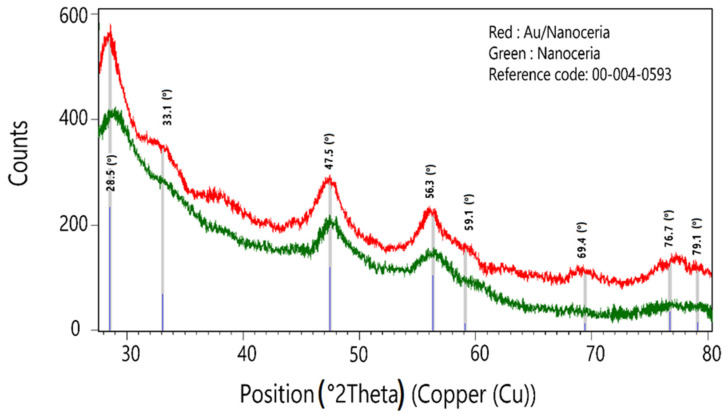
The XRD patterns of nanoceria and Au/nanoceria.

**Figure 2 jfb-14-00332-f002:**
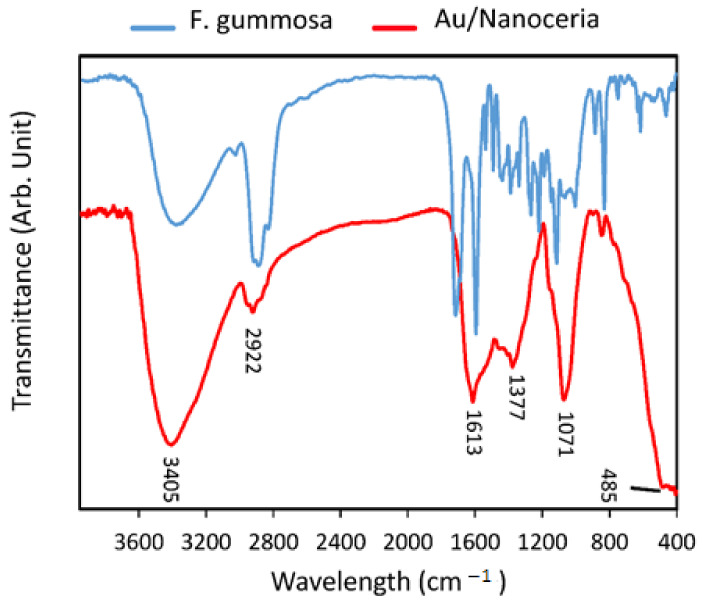
FTIR spectrum of *F. gummosa* and Au/Nanoceria.

**Figure 3 jfb-14-00332-f003:**
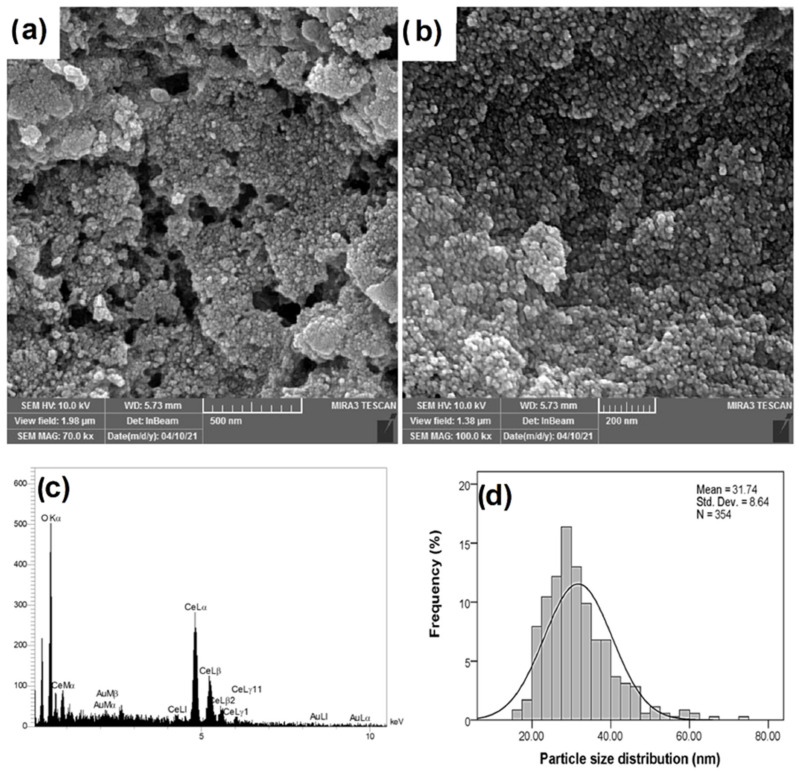
FESEM images (**a**,**b**), EDX analysis (**c**), and particle size distribution (**d**) of Au/nanoceria.

**Figure 4 jfb-14-00332-f004:**
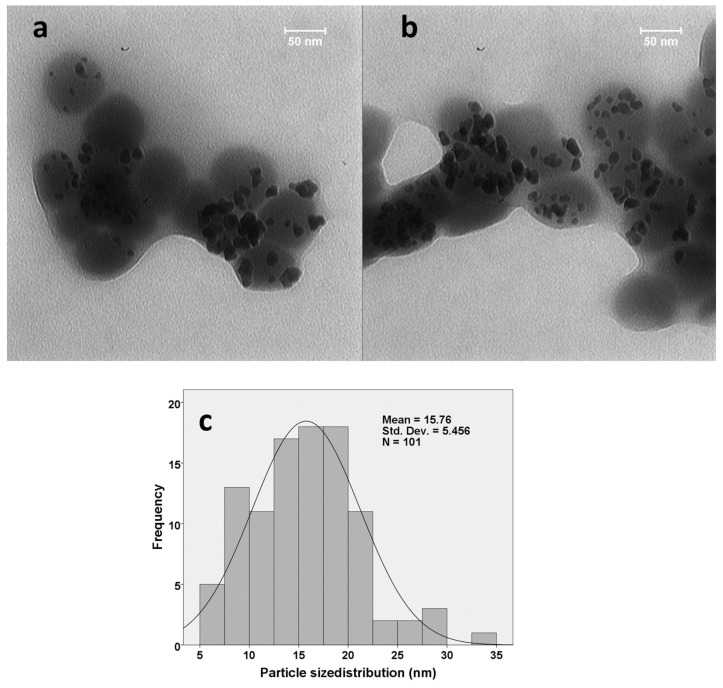
The TEM images (**a**,**b**) and particle size distribution (**c**) of the prepared Au/nanoceria.

**Figure 5 jfb-14-00332-f005:**
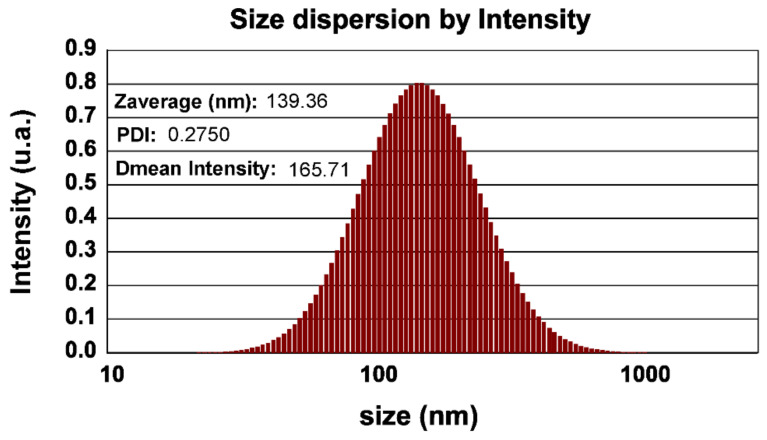
The hydrodynamic sizes of Au/nanoceria.

**Figure 6 jfb-14-00332-f006:**
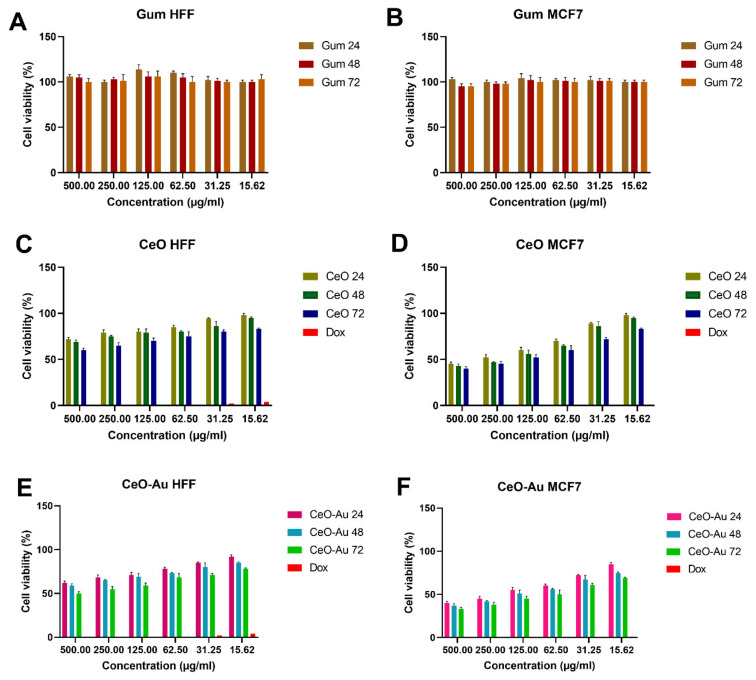
Cellular toxicity effect of biosynthesized Au/nanoceria against HFF cells as a normal cell line, and MCF7 breast cancer cell lines in different doses at 24, 48, and 72 h. (**A**,**B**) Cytotoxicity effects of gum against the HFF and MCF7 cell lines. (**C**,**D**) Cytotoxicity effects of CeO NPs and doxorubicin (Dox) against the HFF and MCF7 cell lines. (**E**,**F**) Cytotoxicity effects of CeO-Au NPs and Dox against the HFF and MCF7 cell lines. Data are represented as the mean ± SD. The test was done in triplicate.

**Table 1 jfb-14-00332-t001:** The comparison of the calculated and experimental 2theta and d-spacing values.

				Calculated Values		Experimental Values (Au/Nanoceria)	
No.	h	k	l	d (A)	2Theta(deg)	d (A)	2Theta(deg)
1	1	1	1	3.12	28.55	3.12	28.59
2	2	0	0	2.71	33.08	2.71	33.09
3	2	2	0	1.91	47.49	1.92	47.43
4	3	1	1	1.63	56.33	1.64	56.12
5	2	2	2	1.56	59.10	1.56	59.22
6	4	0	0	1.35	69.41	1.35	69.68
7	3	3	1	1.24	76.74	1.24	76.47
8	4	2	0	1.21	79.08	1.21	78.79

## Data Availability

Data sharing was not applicable to this article, as no datasets were generated or analyzed during the current study.

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
