# Peer review of "Plant Gel-Mediated Synthesis of Gold-Coated Nanoceria Using Ferula gummosa: Characterization and Estimation of Its Cellular Toxicity toward Breast Cancer Cell Lines"

_jfb, 2023, doi:10.3390/jfb14070332_

Round 1

Reviewer 1 Report

The contribution is original and sound. Yet several bio-engineered publications on nano ceria were published, the originality of the current submission lies within the fact that Ferula gum- 2 mosa gums extract was used as an effective defloculant agent (while it could have been used chelating agent in addition of the Au graphting. The reported results are comprehensive & complementary in support of the various discussions within the manuscript and the deduced conclusions. Hence, it is recommended for publication once the following Precisions (P), Corrections (C) & Recommendations (R) are addressed each & All:

PCR-1:

What is the rationale in choosing Au instead of Ag or Pt  to be anchored onto nanoscaled Ceria:

PCR-2:

 Figure 1  reports the XRD patterns of nanoceria and Au/nanoceria. As one can notice, it is mainly dominated by the nanoCeria  signature . It look like as if the nanogold nanoparticles are amorphous. Is this is the case?

PCR-3: 

It is not clear all on how the synthesis of the nano-Ceria was finalized?

PCR-4:

It would have bee ideal using the natural extract of Ferula gum- 2 mosa gums  as an effective chelating agent. In this regard, the authors are encouraged to refer to the following publication that they should include in their discussion and add them in their reference section:

CeO2 nanoparticles green synthesis by Hibiscus Sabdariffa flower extract: Main physical properties

Thovhogi, N., Diallo, A., Gurib-Fakim, A., Maaza, M., Journal of Alloys and Compounds, 2015, 647, pp. 392–396 

PCR-5:

It is state "Herein, Ferula gummosa gum was used as a capping agent to synthesize nanoceria for 91 the first time. The synthetic method was economic and novel and it was performed at room temperature. Furthermore, the surface of nanoceria was coated with the gold (Au/nanoceria) which was fully characterized and used for cell toxicity effects against 94 breast cancer cell lines (MCF7) and normal cells (HFF)"

What is the nature of the bonding between the Au nanoparticles & the Ceria surface< are they physically or chemically bound. In this regard, the authors are encouraged to refer some publication that they should include in their discussion and add them in their reference section.

PCR-6:

While the anticancer response is effective of the proposed nanocomposite, it is necessary to compare such a response to those published in the literature. In this regard, the authors are encouraged to refer to some publications that they should include in their discussion and add them in their reference section.

Author Response

Reviewer 1

The contribution is original and sound. Yet several bio-engineered publications on nano ceria were published, the originality of the current submission lies within the fact that Ferula gum- 2 mosa gums extract was used as an effective defloculant agent (while it could have been used chelating agent in addition of the Au graphting. The reported results are comprehensive & complementary in support of the various discussions within the manuscript and the deduced conclusions. Hence, it is recommended for publication once the following Precisions (P), Corrections (C) & Recommendations (R) are addressed each & All:

- Many thanks for your efforts to review our manuscript, revising the requested items can definitely improve quality of our manuscript, here are the point by point responses to them:

PCR-1: What is the rationale in choosing Au instead of Ag or Pt  to be anchored onto nanoscaled Ceria:

Response 1: Gold nanoparticles have been studied extensively as potential cancer therapy due to their unique physical and chemical properties. Gold has unique physical and chemical properties that make it well-suited for cancer therapy, and coating nanoparticles with gold provides several advantages compared to other metals. Gold nanoparticles are more biocompatible than silver or platinum, meaning they are less likely to damage cells and tissues. Gold is more stable than silver or platinum, so gold nanoparticles are less likely to degrade or break down in the body, which reduces toxicity. The unique optical properties of gold nanoparticles allow them to be easily visualized using imaging techniques such as CT (computed tomography) or MRI (magnetic resonance imaging), making them useful for diagnostic purposes. Gold nanoparticles can be functionalized with various ligands and antibodies that can bind specifically to cancer cells, allowing for targeted delivery of drugs or other therapeutic agents. Gold nanoparticles can absorb light and convert it into heat, which can be used to kill cancer cells through a process called photothermal therapy. Gold is better as a coating on the surface of nanoparticles in cancer treatment, there are several reasons: (1) Stability: As mentioned earlier, gold is more stable than other metals, so it provides better protection for the nanoparticle core and reduces toxicity. (2) Biocompatibility: Gold is biocompatible, so it is less likely to cause harm to cells and tissues. (3) Surface chemistry: The surface chemistry of gold is well-studied, and it is known to readily form stable covalent bonds with other molecules. This makes it easier to functionalize the surface of gold-coated nanoparticles with targeting ligands or therapeutic agents. (4) Optical properties: Gold-coated nanoparticles have unique optical properties that can be used for imaging and photothermal therapy.

PCR-2: Figure 1 reports the XRD patterns of nanoceria and Au/nanoceria. As one can notice, it is mainly dominated by the nanoCeria signature. It look like as if the nanogold nanoparticles are amorphous. Is this is the case?

Response 2: Thanks a lot. Yes, it is possible.

PCR-3:  It is not clear all on how the synthesis of the nano-Ceria was finalized?

Response 3: XRD (X-ray diffraction) can show the synthesis of cerium oxide (nanoceria) by detecting the peaks or patterns of the crystalline structure of the material. This technique can confirm the formation of cerium oxide and can provide information about the crystal size. FTIR (Fourier Transform Infrared Spectroscopy) can show cerium oxide synthesis by detecting the Ce-O vibration band. The presence of this band indicates the formation of cerium oxide and can provide information about the bonding and structure of the material. Dynamic Light Scattering and FESEM (Field Emission Scanning Electron Microscopy) can show the particle sizes of the synthesized cerium oxide. Dynamic Light Scattering can provide information about the size distribution of the particles in aqueous media, while FESEM can provide high-resolution images of the particles and their morphology in the solid phase. EDAX analysis can show the elemental composition of nanomaterials including cerium oxide and gold. It can detect the presence of various elements in the sample, which can provide information about the purity and composition of the material.

PCR-4: It would have been ideal using the natural extract of Ferula gummosa gums as an effective chelating agent. In this regard, the authors are encouraged to refer to the following publication that they should include in their discussion and add them in their reference section:

CeO2 nanoparticles green synthesis by Hibiscus Sabdariffa flower extract: Main physical properties

Thovhogi, N., Diallo, A., Gurib-Fakim, A., Maaza, M., Journal of Alloys and Compounds, 2015, 647, pp. 392–396

Response 4: Thank you for your suggestion. We used the suggested reference in improving the introduction section.

PCR-5: It is state "Herein, Ferula gummosa gum was used as a capping agent to synthesize nanoceria for the first time. The synthetic method was economic and novel and it was performed at room temperature. Furthermore, the surface of nanoceria was coated with the gold (Au/nanoceria) which was fully characterized and used for cell toxicity effects against 94 breast cancer cell lines (MCF7) and normal cells (HFF)"

What is the nature of the bonding between the Au nanoparticles & the Ceria surface< are they physically or chemically bound. In this regard, the authors are encouraged to refer some publication that they should include in their discussion and add them in their reference section.

Response 5: These are our speculations about the nature of the bonding between the Au nanoparticles & the Ceria surface. We did not find any references specifically explained the nature of bonding between them. But as you mentioned, we think both physical adsorption and chemical bonding are involved to form bonds between the Au coating and the surface of cerium oxide nanoparticles. The van der Waals forces and electrostatic interactions between the Au ions and the cerium oxide surface cause physical adsorption. Meanwhile, chemical bonding occurs as a result of the creation of strong covalent connections between the cerium oxide's surface oxygen atoms and the Au atoms. This bonding is known as chemisorption, and it is responsible for the Au coating's strong adherence to the cerium oxide nanoparticle surface.

PCR-6: While the anticancer response is effective of the proposed nanocomposite, it is necessary to compare such a response to those published in the literature. In this regard, the authors are encouraged to refer to some publications that they should include in their discussion and add them in their reference section.

Response 6: The discussion section s improved according to the opinion of the respected reviewer.

Reviewer 2 Report

The paper entitled “Facile fabrication of gold-coated nanoceria using Ferula gummosa gums: Characterization and estimation of its cellular toxicity toward breast cancer cell lines” by Mousavi-Kouhi et al. reported the efficacy of F. gummosa gum to fabricate Au/nanoceria. The synthesized composite was characterized by FT-IR, XRD, FESEM-EDX, and zeta potential. Moreover, the in-vitro cytotoxicity of the synthesized composite was investigated against the cancer cell line, MCF7. The manuscript needs major revision before being accepted for publication in JFB.   

1-       Line 24, “characterized by using” should be “characterized using”

2-       The hypothesis of the study should be rephrased to be clear.

3-       The introduction section can be improved by referring to the importance of biological synthesis over chemical and physical methods. The following reference can be citing: https://doi.org/10.3390/app122412879

4-       In the “Material and Methods” section, the characterization of composites should be discussed in detail.

5-       Line 228, add the equation used to detect the cellular viability.

6-       In lines 122-123, the author mentioned that the peak at 485 cm-1 is responsible for the formation of Ce-O for nanoceria, whereas this peak also was exist in plant extract, what is the difference?

7-       Please clarify the peaks present in plant extract and their role in the fabrication of Au/nanoceria. Moreover, I recommend adding the FT-IR for nanoceria before the deposition of Au on their surface.

8-        Line 132, please mentioned the number of calculated particles to obtain this value (31.74±8..64 nm).

9-       In figures containing more than one panel, please refer to the number of each panel in the text. For instance, Fig. 3a or b or C or D.

10-    The part of “Result and discussion” needs deep discussion, especially section “2.5”.

11-    Section “3-Conclusion” should be transferred after the “material and method” section. 

Moderate editing of English language is needed

Author Response

The paper entitled “Facile fabrication of gold-coated nanoceria using Ferula gummosa gums: Characterization and estimation of its cellular toxicity toward breast cancer cell lines” by Mousavi-Kouhi et al. reported the efficacy of F. gummosa gum to fabricate Au/nanoceria. The synthesized composite was characterized by FT-IR, XRD, FESEM-EDX, and zeta potential. Moreover, the in-vitro cytotoxicity of the synthesized composite was investigated against the cancer cell line, MCF7. The manuscript needs major revision before being accepted for publication in JFB.

- We would like to thank you for very precise reviewing our manuscript, we do hope that our point by point revision meets your recommended items.

1- Line 24, “characterized by using” should be “characterized using”

Response 1: Done. Thanks a lot for your attention.

2- The hypothesis of the study should be rephrased to be clear.

Response 2: the hypothesis of the study was explained more clearly in the introduction.

3- The introduction section can be improved by referring to the importance of biological synthesis over chemical and physical methods. The following reference can be citing: https://doi.org/10.3390/app122412879

Response 3: The importance of biological synthesis over chemical and physical methods is stated in the Introduction section and this reference is employed in this section.

4- In the “Material and Methods” section, the characterization of composites should be discussed in detail.

Response 4: It is improved.

5- Line 228, add the equation used to detect the cellular viability.

Response 5: It is added. Cell viability (%) = (absorbance of sample/absorbance of cells without treatment) ×100. Thank you for your suggestion.

6- In lines 122-123, the author mentioned that the peak at 485 cm-1 is responsible for the formation of Ce-O for nanoceria, whereas this peak also was exist in plant extract, what is the difference?

Response 6: The difference is the intensity of the peak. In the synthesis of cerium oxide nanoparticles, the intense peak at 480-500 cm-1 indicates a strong Ce-O bond formation, which is expected. On the other hand, in the QSM FTIR, the mentioned peak has a very low intensity comparatively. The overlapping of the high intensity Ce-O peak also masks the low intensity peak of the QSM. Hence, the comparison of the intensities could explain the successful synthesis of nanoceria. Additionally, the washing of nanoparticles also could confirm the presence of negligible amount of F. gummosa, which can clearly affect the intensity of the peaks related to F. gummosa.

7- Please clarify the peaks present in plant extract and their role in the fabrication of Au/nanoceria. Moreover, I recommend adding the FT-IR for nanoceria before the deposition of Au on their surface.

Response 7: The peaks in plant extract clarifies the presence different functional groups in the plant extract that could interact with the cerium ions. In our point of view, there is no need to assign the peaks unless the composition studies was performed. We used FTIR analysis to know if there are any functional groups in the F. gummosa able to stabilize cerium ions. Generally, the cerium ions in the solution are precipitated using hydroxide ion. The reaction conditions can be controlled to tailor the size and morphology of the resulting nanoparticles. The mechanism of nanoparticle formation involves the nucleation and growth of particles from the cerium ions as the NaOH solution was added. The initial nucleation stage involves the formation of small clusters of ions, which then grow through the addition of more base. As the particles grow, they become more stable and are less likely to dissolve back into the solution. The final size and morphology of the nanoparticles are determined by the reaction conditions. The use of F. gummosa as a stabilizing agent for cerium ions can have advantages over water or organic solvents. F. gummosa contain various organic compounds, such as polysaccharides or proteins, which can act as stabilizing agents for the nanoparticles. These organic compounds can influence nucleation and growth rate of the particles. The interaction between the compositional organic compounds and cerium ions can also impart specific properties, such as special morphology, particle size, enhanced bioactivity or improved dispersibility, to the resulting nanoparticles.

8- Line 132, please mentioned the number of calculated particles to obtain this value (31.74±8.64 nm).

Response 8: it is added.

9- In figures containing more than one panel, please refer to the number of each panel in the text. For instance, Fig. 3a or b or C or D.

Response 9: it is added. Thank you for your attention.

10- The part of “Result and discussion” needs deep discussion, especially section “2.5”.

Response 10: This section is modified deeply.

11- Section “3-Conclusion” should be transferred after the “material and method” section. 

Response 11: Done. Thank you very much again for your detailed study.

Reviewer 3 Report

I congratulate the authors on an exciting research. However, there are a few matters for the authors to clarify and to improve the manuscript.

1.     Grammatical issues: Commas before “and” in a list of items

          Italicization of scientific names is missing

2.     SEM and DLS show very different results. Since PI is low, the results of DLS are reliable, and the authors' SEM analysis has been done with oversight. There are clear aggregates visible in the images.  

3.     It is also questionable if gold was actually incorporated into the nanoparticles; the EDX in Figure 3c is not convincing. Authors use “pure” CeO nanoparticles in the toxicity tests but do not show any results from these NPs synthesis and analysis. The comparison between CeO and CeO-Au should be added to convince the reader that gold was added to form a composite.

4.     Figure 5, the main toxicity results, has an insufficient description and data presentation. The authors claim that the nanoceria nanoparticles did not influence the HFF cells, but in Figure 5, cell survival is clearly suppressed. And only at low concentrations, close to 100%, do the cells survive. This section should be rewritten with more details added. What about the statistical differences between different treatments? The authors should add statistical analysis.

5.     Authors claim that the toxicity of nanoceria has not been established, yet some publications show this untrue. For example two reviews: DOI:10.1002/ejic.201500643, or https://doi.org/10.1007/s00018-023-04694-y
With that in mind, the author should compare their results to published literature more thoroughly.   

6.     Clarify the synthesis of nanoceria:

The F. gummosa (0.5 g) was a dry or fresh plant material? If it was plant material, when was it removed?

“After stirring for 24 h, NaOH (0.1) was added dropwise until nanoceria was formed and the supernatant was clear.”

Was the supernatant centrifuged in the next step or the whole solution? 

Author Response

I congratulate the authors on an exciting research. However, there are a few matters for the authors to clarify and to improve the manuscript.

- We would like to thank you for very precise reviewing our manuscript, we do hope that our point by point revision meets your recommended items.

  1. Grammatical issues: Commas before “and” in a list of items

          Italicization of scientific names is missing

Response 1: Done. Thank you for your attention.

  1. SEM and DLS show very different results. Since PI is low, the results of DLS are reliable, and the authors' SEM analysis has been done with oversight. There are clear aggregates visible in the images.

Response 2: The aggregation was explained before. However, we omitted “low” before agglomeration or aggregation word.    

  1. It is also questionable if gold was actually incorporated into the nanoparticles; the EDX in Figure 3c is not convincing. Authors use “pure” CeO nanoparticles in the toxicity tests but do not show any results from these NPs synthesis and analysis. The comparison between CeO and CeO-Au should be added to convince the reader that gold was added to form a composite.

Response 3: Thank you very much for your valuable comment. We used the introduced procedure to coat gold on cerium oxide nanoparticles using ascorbic acid as the reduction agent in a stepwise manner. Therefore, we expected an Au-coating on the nanoceria’s surface was formed. We also centrifuged the mixture and washed the particles to remove any unreacted particles. As explained in the manuscript, energy-dispersive X-ray spectroscopy (EDX) was used to analyze the composition of the nanoparticles. We have mentioned the XRD analysis of CeO and CeO-Au nanoparticles. Our aim was to study the toxicity expression of CeO and CeO-Au nanoparticles on cancer cell lines. Also, as stated in the acknowledgment section, this research received no external funding, and due to the end of our budget, we are not able to further.

  1. Figure 5, the main toxicity results, has an insufficient description and data presentation. The authors claim that the nanoceria nanoparticles did not influence the HFF cells, but in Figure 5, cell survival is clearly suppressed. And only at low concentrations, close to 100%, do the cells survive. This section should be rewritten with more details added. What about the statistical differences between different treatments? The authors should add statistical analysis.

Response 4: It was the aim of this study to compare the toxicity of biosynthesized nanoparticles on cancer cells with the toxicity of nanoparticles on normal cells. It should be noted that nanoparticles have a less toxic effect on normal cells than cancer cells. In other words, these nanoparticles exhibit higher toxicity to cancer cells in a semi-selective manner, indicating their efficiency and safety. Meanwhile, we tested the toxicity of various concentrations and times (24, 48, and 72 hours) and presented a graph of cell viability percentage to demonstrate the effects of concentration-dependent toxicity and time-dependent toxicity. For this test, statistical tests are not necessary.

  1. Authors claim that the toxicity of nanoceria has not been established, yet some publications show this untrue. For example two reviews: DOI:10.1002/ejic.201500643, or https://doi.org/10.1007/s00018-023-04694-y
    With that in mind, the author should compare their results to published literature more thoroughly.   

Response 5: Thank you for your suggestion. It is mentioned that “However, there is still little knowledge about the toxicity, anticancer and antibacterial properties of Au coated nanoceria”. We did not claim that the toxicity of nanoceria has not been established. The cell toxicity section is modified deeply and the suggested sources were used in this section.

  1. Clarify the synthesis of nanoceria:

The F. gummosa (0.5 g) was a dry or fresh plant material?

Response 6: It was a dry material.

7. If it was plant material, when was it removed?

Response 7: After the co-precipitation reaction, the precipitate was centrifuged and washed.

8. “After stirring for 24 h, NaOH (0.1) was added dropwise until nanoceria was formed and the supernatant was clear.” Was the supernatant centrifuged in the next step or the whole solution? 

Response 8: Yes, the supernatant was centrifuged out the solution in the next step. Thank you very much again for your detailed study.

Round 2

Reviewer 1 Report

The revised version has addressed all concerned points.It is recommended for publication as is.

Author Response

Dear reviewer,

Many thanks for taking your time to review our manuscript.

Reviewer 2 Report

The manuscript is suitable for publication in the current form

the manuscript needs moderate English editing

Author Response

(The authors gave the same response as above.)

Reviewer 3 Report

I think the main figure Figure 5 needs more analysis in the text. the authors did not fulfill this section of the review.

I also have issues with the conclusions, that are loosely based on the data presented. 

In the 'conclusions,' the authors claim 
"Gold-caoted nanoceria has demonstrated good anticancer and antibacterial efficiency, which can be used in the design of new nanocomposites for the treatment of cancer or drug delivery purposes" 

there is no evidence of antibacterial efficiency in this manuscript. Please revise.

by the way the word 'coated' is misspelled, what other words are misspelled? 

also, in the 'conclusions' authors state that the NPs killed cancer cells and showed 'low' toxicity against normal cells. Again, going back to Figure 5, the readers need to focus on the Figure and try to analyze the data themselves. what is low toxicity? at 500ppm about 60% of HFF survived vs ~45% MCF7... does that really show low toxicity to normal cells?    

Author Response

First we would appreciate your precise review, we have accordingly revised the manuscript and highlighted the changes in the main file, besides the point-by-point responses are mentioned as follows:

I think the main figure Figure 5 needs more analysis in the text. the authors did not fulfill this section of the review. I also have issues with the conclusions, that are loosely based on the data presented.

Response: In Figure 5, more discussion is presented in the revised version, and also in the conclusion section, we tried to express the content in a concise and useful way. However, if the honorable judge has a specific opinion.

In the 'conclusions,' the authors claim  "Gold-caoted nanoceria has demonstrated good anticancer and antibacterial efficiency, which can be used in the design of new nanocomposites for the treatment of cancer or drug delivery purposes"

there is no evidence of antibacterial efficiency in this manuscript. Please revise.

by the way the word 'coated' is misspelled, what other words are misspelled?

Response: Thank you very much for your attention. These items are corrected.

also, in the 'conclusions' authors state that the NPs killed cancer cells and showed 'low' toxicity against normal cells. Again, going back to Figure 5, the readers need to focus on the Figure and try to analyze the data themselves. what is low toxicity? at 500ppm about 60% of HFF survived vs ~45% MCF7... does that really show low toxicity to normal cells?   

Response: Thanks for this comment. The purpose of this study was to compare the cytotoxicity of NPs against cancer and normal cells, in which the results of this study showed the higher cytotoxicity of NPs against cancer cells. So as the respected reviewer correctly pointed out, the term “low toxicity” was changed to “lower toxicity, “suggesting that the NPs are safer on normal cells.